# Influence of Post-Annealing on the Structural and Nanomechanical Properties of Co Thin Films

**DOI:** 10.3390/mi11020180

**Published:** 2020-02-10

**Authors:** Yeong-Maw Hwang, Cheng-Tang Pan, Ying-Xu Lu, Sheng-Rui Jian, Huang-Wei Chang, Jenh-Yih Juang

**Affiliations:** 1Department of Mechanical and Electro-Mechanical Engineering, National Sun Yat-Sen University, Kaohsiung 804, Taiwan; ymhwang@mail.nsysu.edu.tw (Y.-M.H.); pan@mem.nsysu.edu.tw (C.-T.P.); wind9307086@hotmail.com (Y.-X.L.); 2Department of Materials Science and Engineering, I-Shou University, Kaohsiung 840, Taiwan; 3Department of Physics, National Chung Cheng University, Chia-Yi 621, Taiwan; 4Department of Electrophysics, National Chiao Tung University, Hsinchu 300, Taiwan; jyjuang@g2.nctu.edu.tw

**Keywords:** Co thin films, XRD, pop-in, nanoindentation

## Abstract

The correlations between the microstructure and nanomechanical properties of a series of thermal annealed Co thin films were investigated. The Co thin films were deposited on glass substrates using a magnetron sputtering system at ambient conditions followed by subsequent annealing conducted at various temperatures ranging from 300 °C to 800 °C. The XRD results indicated that for annealing temperature in the ranged from 300 °C to 500 °C, the Co thin films were of single hexagonal close-packed (hcp) phase. Nevertheless, the coexistence of hcp-Co (002) and face-centered cubic (fcc-Co (111)) phases was evidently observed for films annealed at 600 °C. Further increasing the annealing temperature to 700 °C and 800 °C, the films evidently turned into fcc-Co (111). Moreover, significant variations in the hardness and Young’s modulus are observed by continuous stiffness nanoindentation measurement for films annealed at different temperatures. The correlations between structures and properties are discussed.

## 1. Introduction

Cobalt (Co) thin films have been subjected to extensive studies owing to their essential importance in spintronic devices [1,2], such as multilayers giving rise to giant magnetoresistance effect [3,4] and spin valves [5,6]. For practical applications, since the manufacturing and packaging processes are inevitably involving contact loading, it is thus necessary to fully recognize the mechanical characteristics of Co thin films, especially in the nanoscale regime. To this respect, nanoindentation stands out as one of the most efficient tools for obtaining the basic mechanical characteristics, such as the hardness and elastic modulus, at small length scales of various thin films [7,8,9,10] or micro/nano-sized materials [11,12,13,14,15] owing to its advantages of high sensitivity, good resolution and easy operation. In addition, it has been widely recognized that the loading and unloading segments of the nanoindentation load-displacement curves often reflect rich information about the dominant deformation mechanism prevailing in the materials under investigation. For instance, the load-displacement responses provide substantial insights into the onset of plastic deformation, phase transformation, creep resistance or fracture behaviors of the various materials [16,17,18,19,20,21,22,23]. In nanoindentation, it has been ubiquitously identified that the materials often exhibit a characteristic feature called the “pop-in” phenomenon during loading. It manifests as a sudden displacement burst at a nearly constant indentation load, signifying the onset of plastic deformation. In general, the “pop-in” behavior has been interpreted as the manifestations of the dislocation activity [16,17,18], while in some materials it was attributed to either nanoindentation-induced phase transformation [19,20] or to the crack/delamination phenomena of the interface of films and substrates [23,24].

For the nanoindentation of metallic thin films, multiple “pop-in” phenomena are often observed in face-centered cubic (fcc) structured Au and Cu films [25,26], or the hexagonal closed-packed (hcp) structured Ru film [27]. However, no pop-in event was observed previously in Co thin films during nanoindentation [28]. Since during nanoindentation the onset of nanoscale plasticity and subsequent deformation behaviors are strongly influenced by various factors, such as crystal structure [29], temperature [30] or even the radius of indenter tip [31], thus, it is not surprising to see that the consensus on the understanding of the onset of nanoscale plasticity and the plastic deformation mechanisms of Co thin films not only remains largely inconclusive but also even misinterpreted in some cases.

This study focuses on investigating the nanomechanical properties of Co thin films deposited on glass substrates using magnetron sputtering at ambient conditions. In order to manipulate the film microstructures and their influences on the mechanical properties, the films were deliberately annealed at various temperatures ranging from 300 to 800 °C. It is noted that the recently developed density functional theory–based synthetic growth concept can also provide efficient means for analyzing and modeling the chemical composition independently for thin films prepared by vapor-phase deposition techniques, such as the magnetron sputtering used in the present study [32]. The structural features and nanomechanical properties of all Co thin films were characterized by X-ray diffraction (XRD) and nanoindentation, respectively. The hardness and Young’s modulus of Co thin films were obtained by using the continuous stiffness measurements (CSM) nanoindentation technique. The effects of correlations crystalline structure and grain size resulted from annealing processes on the nanomechanical properties of Co thin films are discussed. In addition, we also estimated the number of indentation-induced dislocation loops formed at the initial stage of nanoscale deformation in Co thin films using the classical dislocation theory [33].

## 2. Materials and Methods

The Co thin films investigated in the present study were deposited by magnetron sputtering on glass substrates at room temperature. The base pressure of the deposition chamber was better than 1 × 10^−7^ Torr and the pressure of the working gas (Ar) was kept at 20 mTorr. The thicknesses of Co films were ~300 nm. After deposition, the samples were subjected to rapid thermal annealing at temperatures ranging from 300 to 800 °C for 10 min with a fixed heating rate of 40 °C/sec. During the annealing the pressure of the vacuum chamber was better than 1 × 10^−6^ Torr.

The crystal structure and structural morphology of the Co thin films were analyzed by X-ray diffraction (XRD; Bruker D8, CuKα radiation, λ = 1.54 Å, Billerica, MA, USA). The nanoindentation tests were performed using the Nanoindenter MTS NanoXP^®^ system (MTS Cooperation, Nano Instruments Innovation Center, Oak Ridge, TN, USA) with a continuous stiffness measurement (CSM) technique [34]. A pyramid-shaped Berkovich diamond tip was used as the indenter. Prior to the measurement, the indenter was repeatedly set and reset three times to ensure that the indenter tip was properly in contact with the film surface and any parasitic phenomenon was properly eliminated. Subsequently, the indenter was re-loaded at a strain rate of 0.05 s^−1^ until the indentation depth reaches 50 nm. Moreover, at the peak load, the tip was held for 10 sec to avoid the influence of creep on unloading properties. Finally, the tip was withdrawn with the same strain rate and stopped at about 10% of the peak load. In each test, 20 indentations were performed and each indentation was separated by at least 10 μm to avoid any possible interference from the neighboring indents. It is noted that the surface roughness of the present Co thin films is ranging from 0.26 to 1.03 nm from our AFM analyses (not shown here), which are much smaller than the typical indentation depth (~50 nm) performed in our experiments. Thus, we believe that the surface roughness of the present Co films should not have a discernible effect on the mechanical properties reported here.

The analytic method of Oliver and Pharr [35] was adopted to calculate the hardness and Young’s modulus of all Co thin films. In this analytical scheme, the hardness is defined by dividing the applied indentation load by the projected contact area, i.e., *H* = *P*_max_/*A_p_*. Wherein *A*_p_ is the projected contact area and for a perfectly sharp Berkovich indenter, it is given by *A_p_* = 24.56*h*^2^ with *h* being the true contact depth. Subsequently, the relationship: *S* = 2*βE*_r_(*A*_p_/π)^1/2^ developed by Sneddon [36] was used to derive the elastic modulus of the sample. Here, *S* is the contact stiffness of the material and *β* is a geometric constant with *β* = 1.00 for Berkovich indenter, and *E*_r_ is the reduced elastic modulus. *E*_r_ can be further expressed as:(1)$$\frac{1}{E_{r}}=\left(\frac{1-v_{f}^{2}}{E_{f}}+\frac{1-v_{i}^{2}}{E_{i}}\right),$$
with *ν* being the Poisson’s ratio and the subscripts *i* and *f* being the parameters for the indenter tip and thin films, respectively. For the diamond indenter tip used in the present study, *E*_i_ = 1141 GPa and $v_{i}$ = 0.07 are quoted by Oliver et al.’s research. [35]. Whereas, for the Co films, $v_{f}$ = 0.3 is assumed.

## 3. Results

The XRD patterns of as-deposited and annealed Co thin films are shown in Figure 1a. In this figure, the as-deposited Co thin film exhibits a distinct peak at 2*θ* = 44.7°, which is corresponding to the hcp-Co (002). As the annealing temperature is increased, the intensity of this hcp-Co (002) peak is enhanced progressively, indicating that film is developing a texture along with hcp (002) orientation with increasing annealing temperature. When the annealing temperature is raised to 600 °C, the peak shape appears to become asymmetric, suggesting that new phases might have emerged. Indeed, as shown in Figure 1b, the peak can be de-convoluted into two peaks centering at 2*θ* = 44.5° and *2θ* = 44.7°, corresponding to the peaks of fcc-Co (111) and hcp-Co (002), respectively. A more quantitative analysis indicates that the relative volume percentages of the two coexisting phases are about 24% and 76% for fcc-Co (111) hcp-Co (002), respectively. As the annealing temperature is further raised to 700 °C and 800 °C, it is clear that the diffraction peak has shifted entirely to 2*θ* = 44.5°, indicating that the films have turned completely into (111)-oriented fcc phase, as shown in Figure 1a.

The mean grain sizes (*D*) have been deduced from XRD following the Scherer formula [37], *D* = 0.9*λ*/(*β*cosθ); where *λ* is the wavelength of the X-ray radiation (CuKα, *λ* = 1.5406 Å), *θ* is the Bragg angle and *β* is the angle span of the FWHM of the characteristic peak. The estimated mean grain sizes of Co thin films are 24, 28, 39, 45, 37, 32 and 34 nm for the as-deposited film and those annealed at 300 °C, 400 °C, 500 °C, 600 °C, 700 °C and 800 °C, respectively. The grain size of hcp-structured Co thin films (ranged from 300 °C to 500 °C) increased slightly with annealing temperature and the film’s crystallinities are also improved because of the annealing. Whereas, for films being annealed at 600 °C, a decrease in the average grain size is evident, which presumably is originated from the average effect of the original hcp-structured grains and the newly nucleated fcc-structured grains. At the annealing temperature of 700 °C and 800 °C, one sees that the grain size starts to increase with the annealing temperature again. All of these observations are, in fact, in line with conventional nucleation and growth scenarios. Namely, higher annealing temperature favors the thermally activated atomic diffusion, which, in turn, not only repairs the defects existing within the grains, but also facilitates the coalescence of adjacent grains. As a result, the overall film crystallinity is significantly improved.

Figure 2 shows the typical CSM load-displacement curves for the as-deposited Co thin film and for those being rapidly annealed at various temperatures. All the curves clearly exhibit several displacement bursts on the loading curve, known as the characteristic multiple “pop-in” phenomena. The multiple pop-ins can be regarded as the manifestation of sudden activities of dislocations [19,38], giving rise to the seemingly discontinuous plastic deformation in the course of nanoindentation. It is noted that the sudden burst displacement (*d*_pop-in_) widens progressively for films treated with increasing annealing temperatures. The width of the pop-in displacement evidently increased from 1.62 nm to 7.64 nm when the annealing temperature was increased from 300 °C to 800 °C. This interesting phenomenon can be attributed to the following two possible factors. (1) Higher annealing temperatures may have removed the defective lattice more completely. Thus, the dislocations can be glide over a longer distance without encountering the hindrance from lattice defects. (2) The newly formed fcc-phase is having more dislocation slip systems than that of the hcp phase. As a result, for films being annealed at higher temperatures, the activity of indentation-induced dislocations is significantly enhanced. This is also consistent with a significant reduction in the critical loading (*P*_c_) for the first pop-in event to take place, as seen in Figure 2. This behavior also can be observed in the experimental results of Pt (100) [39] and the multiscale modeling of Al [40] during nanoindentation.

The values of hardness and Young’s modulus of the Co thin films as a function of annealing temperature are plotted in Figure 3a. It is evident that both the hardness and Young’s modulus decrease significantly with increasing annealing temperature. The nanoindentation results showed that the hardness of the as-deposited Co thin film was about 9.4 GPa and hardness decreased to about 8.2 GPa when the thin film was annealed at 300 °C, which may have resulted from the relaxation of residual stress driven by the annealing process [41]. Moreover, the reduction of the film hardness of the hcp-structured Co thin films appears to coincide with the increasing grain size when the annealing temperature is increased from 300 to 500 °C in. Similarly, for the fcc-structured Co thin films (annealed at 700–800 °C) the same trend was followed. As displayed in Figure 3b, for both phases, the respective film hardness as a function of grain size follows closely to the well-known Hall–Petch relationship [42]. Since within the context of the Hall–Petch effect the grain boundary hindering dislocation movement is playing the primary role, the apparent multiple “pop-in” events observed in load-displacement curves should have been linked to abrupt plastic flow generated by massive dislocation activities. This also explains why the pop-in events became more pronounced during nanoindentation for films being annealed at higher temperatures. Indeed, the fits to the Hall–Petch equation for the hcp- and fcc-structured Co films (see the dashed lines and fitting results shown in Figure 3b) evidently show that, although the Hall–Petch equation describes well for both phases, the former is having a larger hardness than the latter, presumably due to the lesser slip systems available in the hcp phase.

The average values of hardness and Young’s modulus of all Co thin films obtained are listed in Table 1. The values of hardness and Young’s modulus obtained in this work are significantly larger than those reported by Koumoulos et al. [29], wherein the hardness and Young’s modulus are 5 GPa and 64 GPa, respectively. Moreover, in their experiments, no pop-in event was observed. These discrepancies could have resulted from the fact that in their experiments the indentation depth is over the 30% depth/thickness criterion suggested by Li et al. [16]. Consequently, the obtained experimental values may involve the parasitic effects from the substrate, which can blur the intrinsic properties of the films under investigation. In the present case, since the substrate effects have already been taking into account by using the CSM mode nanoindentation tests, therefore the difference of hardness and Young’s modulus values obtained here can be largely attributed to the microstructural changes triggered by the thermal annealing.

Another noticeable feature in Figure 2 is that there is no discontinuity observed in the unloading curve, namely the so-called “pop-out” event is absent in the present cases. In general, the “pop-out” event is a signature of pressure-induced phase transition and has been observed in Si single-crystals [20] and Ge thin films [43]. The absence of this event indicates that there might not be any phase transition involved in this work. To this respect, Yoo et al. [44] investigated the pressure-induced structural phase transition in Co using high-pressure XRD experiments and found that the hcp Co undergoes a phase transition to the fcc phase in the external pressure range of 100–150 GPa, which is much larger than the apparent pressure exerted by the indenter tip here. Taking the largest load of ~0.6 mN, the indentation depth of ~50 nm from the load-displacement curve of the as-deposited film displayed in Figure 2, and *A*_p_ = 24.56*h*^2^, the estimated pressure under the Berkovich indenter tip is about 9 GPa. Consequently, it is plausible to assert that the deformation behavior/mechanism in Co thin film is mainly dominated by massive dislocation nucleation and propagation. 

Having established the intimate correlations between the first pop-in event appearing in the loading segment and the dislocation activities triggered by indentation, we further proceed to derive the prominent information about the critical shear stress (*τ*_max_) and the energy associated with the nucleation of dislocation loops. According to the analytical model proposed by Johnson [45], *τ*_max_ is given by:(2)$$\tau_{max}=\frac{0.31}{\pi }{\left(\frac{6P_{c}E_{r}^{2}}{R^{2}}\right)}^{1/3},$$
where *R* and *P*_c_ are the radii of the indenter tip and the critical load for triggering the dislocation movement. If we further assume that the triggered plastic deformation is manifested by generations of circular dislocation loops, then the free energy associated with a circular dislocation loop of the radius (*r*) can be written as:(3)$$F=2\pi rW-\pi r^{2}b\tau_{e},$$
where *W* is the line energy of dislocation loop, *b* is the magnitude of Burgers vector and $\tau_{e}$ is the external shear stress. The first term can be regarded as the energy cost for creating a dislocation loop in an initially defect-free lattice. Whereas, the second term of Equation (3) describes the relief of lattice strain energy resulting from the work done by expanding the dislocation loop over a displacement of one Burgers vector under the applied shear stress. The line energy of a circular dislocation is mainly due to lattice strain caused by the existence of the dislocation and is given by Hirth et al.’s research [33]:(4)$$W=\frac{2-v_{f}}{2\left(1-v_{f}\right)}\cdot \frac{\mu b^{2}}{4\pi }\cdot \left(ln\frac{4r}{r_{core}}-2\right),$$
where μ is the shear modulus and *r*_core_ is the radius of the dislocation core. Similar to most of the nucleation and growth processes, the free energy *F* of the system has a maximum value when the dislocation loop has the critical size of *r*_c_. This maximum energy is expected to decrease with the increasing load. Consequently, the pop-in event, i.e., the simultaneous formation of homogeneous circular dislocation loops, becomes possible without the aids of thermal energy at *F* = 0. Using this condition and *dF*/*d r*= 0 for a maximum are fulfilled if: *τ*_c_ = 2*W*/*br* and *r*_c_ = (*r*_core_·e^3^)/4, wherein the value of the critical resolved shear stress *τ*_c_ is taken as half of *τ*_max_ [46]. Taking two conditions for calculating, we can obtain hcp phase (for annealed@500 °C case, *r*_core_ ≈ 0.35 nm and *r*_c_ = 1.75 nm) and fcc phase (for annealed@800 °C case, *r*_core_ ≈ 0.43 nm and *r*_c_ = 2.15 nm) of Co thin films.

On the other hand, the number of dislocation loops generated in the pop-in can be calculated from the associated work-done, *W*_p_, during nanoindentation. *W*_p_ is approximated as the product of *P*_c_ and *d*_pop-in_, as listed in Table 1. For the hcp Co thin film (annealed at 500 °C), *W*_p_ ≈ 1.3 × 10^−1^^3^ Nm is obtained, implying that ~4.6 × 10^4^ dislocation loops with the size of critical diameter are generated during the pop-in event. Likewise, ~9.4 × 10^4^ dislocation loops are formed (as *W*_p_ ≈ 2.7 × 10^−1^^3^ Nm) for fcc Co thin film (annealed at 800 °C). These numbers are relatively low and are in agreement with the scenario of homogeneous dislocation nucleation, instead of activating the collective motion of pre-existing dislocations [47].

## 4. Conclusions

In conclusion, the microstructural and nanomechanical characterizations of annealed Co thin films are investigated by XRD and nanoindentation, respectively. The XRD results indicated that the structure of Co thin films exhibited the coexistence phases of hcp-Co (002) and fcc-Co (111) at the annealing temperature of 600 °C. In addition, the nanoindentation results indicated that the hardness of Co thin films decreased from 8.2 ± 0.1 GPa to 4.8 ± 0.2 GPa with the increased annealing temperature (300 °C → 800 °C); likewise Young’s modulus decreased from 124.4 ± 12.4 GPa to 92.5 ± 8.2 GPa. Furthermore, the energetic estimation indicated that the number of dislocation loops of fcc-Co is twice more than hcp-Co for the first pop-in event, presumably due to the more slip systems in the fcc structure.

## Figures and Tables

**Figure 1 micromachines-11-00180-f001:**
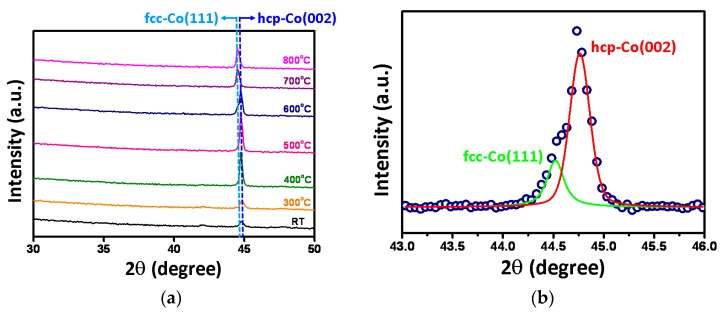
(**a**) X-ray diffraction (XRD) patterns of the as-deposited Co thin film, and those being annealed at temperatures of 300 °C–800 °C. (**b**) The fitting data of XRD result for 600 °C-annealed Co thin film, indicating that the coexistence of hcp-Co (002) and fcc-Co (111) peaks, locating at 2*θ* = 44.7° and 2*θ* = 44.5°, respectively.

**Figure 2 micromachines-11-00180-f002:**
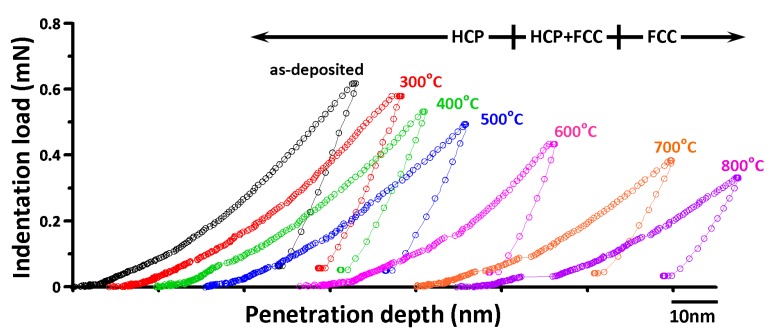
The continuous stiffness measurement (CSM) load-displacement curves of the as-deposited Co film and those being annealed at various temperatures, ranging from 300 °C to 800 °C.

**Figure 3 micromachines-11-00180-f003:**
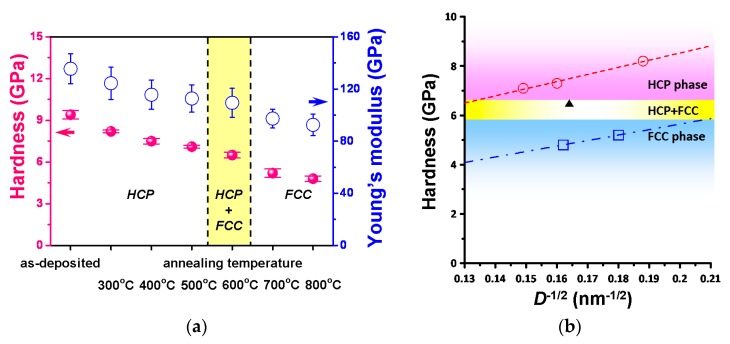
(**a**) Hardness and Young’s modulus of Co films as a function of annealing temperature of the as-deposited and annealed Co thin films. (**b**) The hardness as a function of grain size, showing similar behaviors for both the hcp- and fcc-structure Co thin films. The dash lines are the fits to the Hall–Petch equation: *H*(*D*) = *H*_0_ + *kD*^−1/2^, with *H*(*D*) = 28.9*D*^−1/2^ + 2.7 and *H*(*D*) = 22.2*D*^−1/2^ + 1.2 for hcp- and fcc-structured Co thin films, respectively.

**Table 1 micromachines-11-00180-t001:** The structural properties and values of *D*, *H*, *E*_f_, *P*_c_, *d*_pop-in_ and *τ*_max_ of Co thin films in this study. The results of Au thin films reported previously are also listed for comparison.

Co thin films	structure	*D* (nm)	*H* (GPa)	*E*_f_ (GPa)	*P*_c_ (mN)	*d*_pop-in_ (nm)	*τ*_max_ (GPa)
as-deposited	hcp	24	9.4 ± 0.3	135.5 ± 11.5	0.102	1.04	3.1
annealed@300 °C	hcp	28	8.2 ± 0.1	124.4 ± 12.4	0.088	1.62	2.7
annealed@400 °C	hcp	39	7.5 ± 0.2	115.6 ± 11.3	0.078	1.65	2.5
annealed@500 °C	hcp	45	7.1 ± 0.1	112.7 ± 10.5	0.075	1.76	2.4
annealed@600 °C	Hcp + fcc	37	6.5 ± 0.2	109.4 ± 11.1	0.062	1.84	2.1
annealed@700 °C	fcc	32	5.2 ± 0.3	97.3 ± 7.2	0.058	2.32	1.7
annealed@800 °C	fcc	38	4.8 ± 0.2	92.5 ± 8.2	0.036	7.64	1.6
Au thin films [25]	fcc	–	1.07–2.79	68–99	~0.06–0.10	~5–15	~2.5–4.7

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
