# Peer review of "Influence of Post-Annealing on the Structural and Nanomechanical Properties of Co Thin Films"

_micromachines, 2020, doi:10.3390/mi11020180_

Round 1

Reviewer 1 Report

This manuscript is dedicated to the influence of post-annealing on the structural and nanomechanical properties of Co thin films.

The present work and corresponding result discussion are well done and provide a valuable angle of understanding of the topic. It is also an easy-to-follow and informative read.

In order to deserve publishing, it needs some relatively minor revision:

1: Introduction is written well. However, it should be mentioned that efficient DFT-based methods such as Synthetic Growth Concept (J. Phys. Chem. C 2014, 118, 6514−6521) are efficient in analyzing and modelling thin films deposited by magnetron sputtering independently on the chemical composition of these films.

2: Have the authors considered a concise inclusion of a study related to surface roughness and its possible impact on films’ nanomechanical properties?

3: The section dedicated to H, Ef, Pc, dpop-in and τmax of present Co thin films should compare these results (at least) to other Co-based thin films and/or to similar transition metal films (Ru, or Au-based).

4: Conclusions should not contain numbered items. Also, in the conclusions any foreseen applications should be briefly mentioned.

Author Response

Please see the attached revised report, thanks!

Jian

Reviewer 2 Report

The manuscript presents original research that expands our knowledge in the field of mechanical properties of metal thin films, particularly Co thin films. In general, the manuscript written by acceptable English and may be published as is.

Author Response

(The authors gave the same response as above.)
